



# Numerical Models for Monitoring and Forecasting Sea Ice: a short description of present status

Laurent Bertino[1], Patrick Heimbach[2], Ed Blockley[3], Einar Ólason[1]

[1]Nansen Environmental and Remote Sensing Center, Bergen, Norway
[2]Oden Institute for Computational Engineering and Sciences, The University of Texas at Austin, Austin, TX, United States
[3]Met Office Hadley Centre, Exeter, United Kingdom

*Correspondence to*: Laurent Bertino (laurent.bertino@nersc.no)

**Abstract.** The severe changes in climate resulting in the polar oceans getting warmer – with drastic consequences to their physical, biogeochemical and biological state – require forecasting systems that can accurately simulate and skilfully predict the state of the ice cover and its temporal evolution. Sea-ice processes significantly impact ocean circulation, water mass formation and modifications, and air-sea fluxes. They comprise vertical processes, mainly related to thermodynamics, and horizontal ones, due to internal sea ice mechanics and motion. We provide an overview on how these processes can be modelled and how operational systems are working, in combination with data assimilation techniques, to enhance accuracy and reliability. We also emphasize the need for advancing research on improving such numerical techniques by highlighting currents limits and ways forward.

## 1 Introduction

The main objective of an operational sea-ice forecasting system is to provide users with a reliable estimate of the state of the ice cover and its temporal evolution. To meet this goal the system needs to be coupled to, or use data from, ocean and atmosphere forecasting systems. Some form of data assimilation is also required to provide the model with the best possible starting position, accounting for the chaotic nature of the atmosphere-ocean-ice system. Users of sea-ice forecasting systems can either be ship captains operating in the polar regions or intermediate service providers. With a changing climate and warming polar oceans, the number of stakeholders interested in operating in ice infested waters is growing.

Sea-ice processes have a profound importance for the ocean circulation and water mass modifications, so that ocean models of the polar regions are always coupled to a sea-ice model, both for operational forecasting and climate projection purposes. Sea ice models have their origin in the climate modeling community in the 1970's and were subsequently part of the ocean general circulation model. They have since then evolved to provide sea ice forecasts in their own right and have been made modular to avoid being bound to a given choice of physical ocean model (Blockley et al., 2020). Sea ice observations from satellites are assimilated in the prediction systems (Buehner et al., 2017). This chapter gives a summary of the short-term (up to 10 days) sea ice forecasting systems for the polar regions.





## 2 Overview of processes in sea ice

The physical processes simulated by sea-ice models are commonly split into two: vertical processes, related to thermodynamic growth and melt, and mechanical and dynamical processes influencing the horizontal movement of ice. This dynamic-thermodynamic separation has practical advantages for computations.

### 2.1 Thermodynamics

The ocean can freeze in different phases of sea ice, starting with frazil crystals and their conglomerates into a liquid mush referred to as grease ice, then pancake ice in the presence of waves or slush when the waves flood the snow (Wadhams 2000). Slush, grease, pancakes and ice may sound like a perfect birthday party, until you realize that there is also salt in the ice (Feltham et al., 2006, de la Rosa et al., (2011), Jutras et al., (2016)). The latter will slowly flow out of the ice through brine channels, but usually after its multi-year birthday party (e.g., Notz and Worster, 2009). Once a layer of ice has formed on the surface of the ocean, new ice is mostly formed from below as crystals moving upward from the ocean mixed layer affix to the base of the ice in a process known as 'congelation growth'. Sea ice also freezes laterally within open leads and between ice floes. Snow accumulates on top of the sea ice and forms an efficient thermal insulator as well as a white coating that reflects solar radiation back to the atmosphere. A smaller amount of snow-ice comes from compacted snow above the ice. The insulating effect of snow inhibits both sea ice growth in early winter and sea ice melt in late winter (Bigdeli et al., 2020).

When summertime approaches, the snow melts first, and forms melt ponds at the surface of the ice. These dark ponds absorb more solar radiation and enhance the summer melt.

The sea ice itself can be seen as an insulating layer between the ocean and the atmosphere, with thick ice a better insulator than thin ice.

### 2.2 Mechanics

Sea ice deforms under the action of winds and currents. Their surface drag accumulated over hundreds of kilometers of sea ice results in formidable forces able to crack open the thickest ice or pile it up into pressure ridges, cracks, leads and ridges in what are called linear kinematic features of sea ice. First-year ice can become about 1 meter thick while multi-year ice is more often deformed via compressive stresses and can easily reach 2 meters or above. The convergence of ice is a major threat to navigation and only a few ice-strengthened vessels or icebreakers are designed to withstand such forces. The deformation of sea ice has been measured by drifting buoys and satellite data and scaling laws have revealed multi-fractal properties (Weiss and Marsan, 2004) and power-law behavior (Weiss et al., 2009).

Waves formed in the open ocean will often reach the ice and attenuate within the ice pack, flexing and occasionally breaking the ice into smaller floes along the way. Smaller ice floes offer more reflecting edges and are more efficient at attenuating waves. This represents a negative feedback in the wave-ice interactions (Squire, 2020). This equilibrium results in a wave-broken marginal ice zone (MIZ) which is typically 100 km wide in the Arctic but can reach 1000 km in the Southern Ocean



where waves are bigger and the ice thinner. Sea ice can also be submerged by waves, making it saltier at the surface. Wave effects enhance the lateral melting of ice during summer, but also enhance its freezing during winter.

## 2.3 Biogeochemistry

There is life in sea ice, not only the occasional seal innocently sunbathing as the polar bear lurks around, but as a dense activity under the sea ice following the growth of red ice algae (Duarte et al., 2017). The availability of light below the ice and the size of brine channels determines the growth of algae and the peculiar ecosystem that depends on them (Arrigo, 2014).

## 2.4 Numerical models

Many sea-ice models are complex community codes, simulating the dynamical properties (the constitutive law or rheology) and the thermodynamics of sea ice. The most widespread rheological model of sea ice is the Viscous-Plastic model, often met in the Elastic-Viscous-Plastic (EVP) form which is more efficient for massively parallel computing. One or the other is implemented in the Community sea Ice CodE (CICE), the Sea Ice modelling Integrated Initiative (SI$^3$), the Louvain-la-Neuve sea Ice Model (LIM), the MIT general circulation model (MITgcm), and GFDL's Sea Ice Simulator (SIS2). The previous models all use an Eulerian model grid, but a recent code, the neXt generation Sea Ice Model (neXtSIM) has adopted an adaptive Lagrangian mesh, as well as a more recent Brittle-Bingham Maxwell rheology (Ólason et al., 2022) that exhibits linear features of sea ice deformations apparent in Figure 1. All recent sea-ice models are multi-category models and thus explicitly simulate an ice thickness distribution. They also include a sea-ice age tracer and can thus predict areas of FYI and MYI. Their use in operational forecasts is indicated in Table 1.

The above ocean and sea-ice models are coupled via advanced software (OASIS, ESMF, CCSM) that make them modular, but some ocean models come with an integrated sea ice model, for example the MITgcm, the MOM, the HBM and the FESOM2 codes. The latter is using finite volume (Danilov et al., 2017).

## 2.5 Data assimilation

The most important step to initialize a forecast is to assimilate the latest available observations into a numerical model. Some of the most important observations are available in near-real time with sea-ice concentration, thickness, and motions, but feeding them into the model is a delicate matter (Bertino and Holland, 2017; Buehner et al., 2017). Unobserved variables as well as the ocean properties below the ice must be estimated by multivariate update because of the complex processes both within the sea ice and between the ice and ocean. The irregular observational sampling also requires a flow-dependent spatial interpolation. Operational centers run numerical models and data assimilation codes on dedicated High-Performance Computers (HPC).



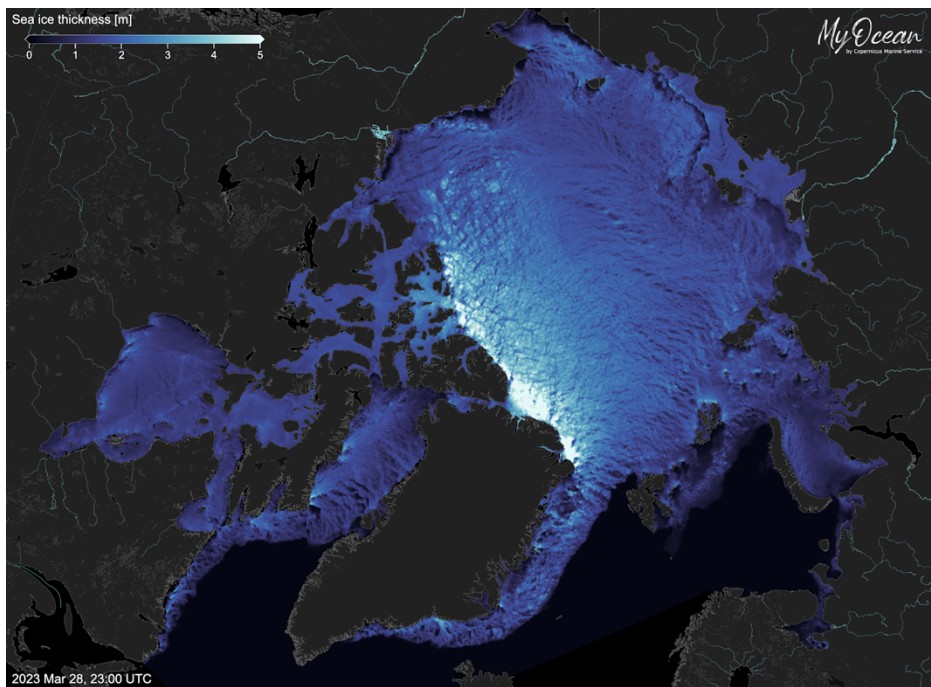

**Figure 1: Example of sea ice thickness analysis from the neXtSIM-F system, visualisation from the Copernicus Marine Services (http://marine.copernicus.eu).**

**Table 1: List of present-day short-term Global and Arctic forecast systems including specification of spatial resolution, sea ice model, assimilation method, variables and website. Sea ice variables are SIC concentration, SIT thickness, SIUV motions, SIALB albedo, SNOW snow depth, SIAGE ice age. Note that the output spatial resolution may differ from the native resolution. Baltic forecasting systems are omitted for brevity. Ocean data assimilated are also omitted. * Output interpolated to 9 km. ** VENUS is deployed on demand.**

| Area | Country | System name | Resolution at NP (km) | Sea ice Model | Assimilation (method and sea ice data) | Variables distributed | Website |
|---|---|---|---|---|---|---|---|
| Arctic | P.R. China | ArcIOPS | 18 km | MITgcm | LESTKF SIC, SIT | SIC, SID, SIT | http://www.oceanguide.org.cn/IceIndexHome/ThicknessIce |
| Global | USA | RTOFS | 3.5 km | CICE5 | 3DVAR SIC | SIC, SIT, SIUV | https://polar.ncep.noaa.gov/global/ |
| Arctic | Norway | TOPAZ 5 | 6.25 km | CICE5 | EnKF SIC, SIUV, SIT | SIC, SIT, SIUV. SNOW, SIALB, SIAGE | https://marine.copernicus.eu/ |



| | | | | | | | |
|---|---|---|---|---|---|---|---|
| Arctic | Norway | neXtSIM-F | 3km (output) | neXtSIM | Nudging SIC | SIC, SIT, SIUV, SNOW, SIALB, SIAGE | https://marine.copernicus.eu/ |
| Global | France | MOi | 3.5 km | LIM2 | SEEK SIC | SIC, SIT, SIUV | https://marine.copernicus.eu/ |
| Global | Canada | GIOPS | 12 km | CICE4 | 3DVAR SIC | | CONCEPTS - Science.gc.ca |
| Arctic | Canada | RIOPS | 3.5 km | CICE4 | 3DVAR SIC | | https://science.gc.ca/eic/site/063.nsf/eng/h_97620.html |
| Global | USA | GOFS3.1 | 3.5 km | CICE4 | 3DVAR SIC | SIC, SIT, SIUV | https://www7320.nrlssc.navy.mil/GLBhycomcice1-12 |
| Global | Europe | ECMWF | 12 km | LIM2 | 3DVAR SIC | SIC, SIT | https://www.ecmwf.int/en/forecasts/datasets/set-i |
| Arctic | Denmark | DMI | 10 km | CICE4 | Nudging SIC | | http://ocean.dmi.dk/models/hycom.uk.php |
| Global | UK | Met Office coupled DA | 12 km | CICE5 | 3DVAR SIC | SIC, SIT, SIUV | https://marine.copernicus.eu/ |
| Global | UK | Met Office FOAM | 3.5 km | CICE5 | 3DVAR SIC | | |
| Arctic ** | Japan | VENUS | 2.5km | IcePOM | N/A | SIC, SIT | https://ads.nipr.ac.jp/venus.mirai/#/mirai |

The data assimilation methods in operation are most often the 3D-variational (3DVAR) method (Tonani et al., 2015; Waters et al., 2015; Mogensen et al., 2012; Hebert et al., 2015; Smith et al., 2016; Usui et al., 2006), assimilating sea-ice concentration and more recently sea-ice thickness (Mignac et al. 2022). The 4DVAR method is not presently used in operational forecasts but can provide long-term optimized model trajectories that are fully consistent with the model equations (Nguyen et al., 2021). The Ensemble Kalman Filter (EnKF) is also used in the TOPAZ system to assimilate concentrations, thickness, and motion

vectors (Xie et al., 2017) and has been tested with neXtSIM (Cheng et al., 2023) although a cheaper nudging is used operationally (Williams et al., 2021). The EnKF does not intrude in the model software and the resulting forecast system is



modular. Even though operational centers use the state of the art with respect to sea-ice data assimilation, they are still inaccurate in locating the ice edge (about 40 km at analysis time, Carrières et al., 2017), even less accurate in locating the boundary between first-year and multi-year Ice (200 km errors rather than 40 km).

With improved observational data coverage and increased computational power, rapid improvements in sea ice modelling and forecasting capabilities are expected in the coming decade. One research thrust concerns the modelling of sea ice as individual floes (e.g., Horvat et al., 2022). A second thrust concerns the development of improved and faster numerical solvers (e.g., Shih et al., 2023). Finally, machine learning approaches are flourishing, which seek to develop fast, surrogate modelling and forecasting capabilities (e.g., Hoffman et al., 2023, Durand et al., 2024, Gregory et al., 2024).

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

**Competing interests**

The contact author has declared that none of the authors has any competing interests.

**Data and/or code availability**

Data used in Figure 1 is freely available from https://marine.copernicus.eu

**Authors contribution**

LB prepared the manuscript with contribution from all co-authors.

**Acknowledgements**

LB acknowledges financial support from the European Union's Horizon Europe project ACCIBERG (grant agreement No. 101081568).