# Peer review of "Numerical Models for Monitoring and Forecasting Sea Ice: a short"

_State of the Planet, 2024_

## Author Response (AR1)

Review of Numerical Models for Monitoring and Forecasting Sea Ice: a short description of present status

**RC1**: 'Comment on sp-2024-24', Anonymous Referee #1, 19 Oct 2024 reply

General comments:

This paper provides an overview of sea ice process, modeling, and short-term forecast by referring to previous literature including the most recent papers. It would be very helpful for general readers who are not familiar with the sea ice modeling and forecast but are interested in their product. Since this paper focuses on the general description of the past studies, there are no new values to be added in the sea ice research community. Overall, the paper is well written and organized with some evidence, but there are a few more things to be added in the paper which would help further improve the understanding of the sea ice process, modeling, and forecast. Below are major and more specific comments on this paper.

1. Biogeochemistry (L65-68)

The authors mentioned only about the sea animals and algae around the sea ice, but the sea ice also plays an important role in the exchange of natural and anthropogenic gases such as carbon dioxides and aerosols that are crucial sources of nutrients for the phytoplankton and other  sea life below the sea ice. Could the authors expand the section a bit more by adding a few more sentences on the sea ice role in the biogeochemical cycle (e.g., carbon and nutrients)?

*Thanks for bringing more issues to our attention. The following points are added:*

- *The algae will find nutrients in the sea ice, some will be trapped in the ice during freezing, providing a sheltered food store for micro-organisms and then later ejected to the ocean through brine channels (Lund-Hansen et al. 2024).*
- *Sea ice carries sediments while drifting from the shallow shelf seas to the central Arctic, together with nutrients, various biological materials and occasionally pollutants (Krumpen et al. 2019).*
- *Sea ice acts as a lid preventing the exchange of greenhouse gases between ocean and atmosphere, but the sea ice also holds its own carbon pump accounting for 30% of the Carbon uptake in the Arctic (Richaud et al. 2023).*

2. Model bias and further improvement (L107-L114)

At the end of this paper, the authors discussed the model bias in forecasting the sea ice edge and boundary between the first and multi-year ice, but what are the underlying causes of these model biases (e.g., model physics, resolutions, ensemble members, data assimilation techniques, and/or observation)? Also, what measures can the sea ice modeling community undertake the most to reduce the forecast errors? The authors suggested two research thrusts

in the end, but I could not find the link to these errors, wondering how these suggestions can help resolve the model biases.

*Thanks for keeping our feet on the ground, we have tried to improve the logic and make this section more conclusive.*

> *Biases in sea ice area coverage arise from multiple sources, primarily from biased ocean and atmospheric boundary conditions, but also intrinsic biases of the sea ice model itself. These biases interact with each other in complex ways (feedback loops or cancellation of errors). Data assimilation methods rely on unbiasedness assumptions and do not remove biases entirely, often transferring them to unobserved variables. Short of a complete observing network, there are ongoing efforts in improving sea ice models that we believe can reduce biases, provided that incoming biases from new ocean and atmospheric models are also reducing.*

> *[...]*

> *Sea-ice exists at the boundary between the atmosphere and ocean, so sea-ice forecasts depend on accurate atmosphere, ocean, and even wave forecasts. Improving those is, therefore, very important for improving sea-ice forecasts. We see fully coupled atmosphere-ocean-wave-ice models with fully coupled data assimilation as a vital long-term goal for sea-ice forecasting systems.*

> *Even though every improvement of the atmosphere, ice or ocean models is welcome, they require time-consuming rounds of testing in forced and coupled models. In the meantime, post-processing techniques, now aided by machine learning, are a novelty in sea ice forecasting (Parleme et al. 2021, 2023) and reanalysis (Edel et al. 2025).*

Specific comments:

L20: To meet this goal,

*OK.*

L39, 41: "birthday party" is a bit narrative. Is it a well-accepted expression in the scientific community?

*It is not, this is an original attempt to catch the readers' attention. We expect readers outside of the sea ice community and hope that some light humour will stimulate their long-term memory processes. There are other basic emotions such as sadness, fear or anger that we will not attempt to trigger.*

L40: This process is called "brine rejection", so you may add this word in the sentence.

*Thanks, added.*

L60: I am a bit confused. Does it mean a positive feedback, because the waves get amplified with smaller ice floes and generate more ice floes with smaller scales.

*No, it is a negative feedback: the waves are scattered by more numerous ice-ocean edges of smaller floes and are attenuated. Clarified in the text.*

Figure 1: Do you have any satellite observation map to validate the model simulation? This would help readers to understand how reasonably the model reproduces the sea ice thickness.

*OK, adding CS2SMOS. Note that since the original paper submission, the neXtSIM-F forecast has included the assimilation of CS2SMOS. The figures are both updated and the caption acknowledges that the observation is not independent.*

Table 1: "SIUV motions" should be "SIUV velocities". What is "SID"? Also, remove "**" in the area of the IcePOM in the table.

*Thanks for the corrections. SID is Sea ice Drift, replaced by SIUV in the new text.*

*Stars are footnotes to the captions and are kept.*

**Citation**: https://doi.org/10.5194/sp-2024-24-RC1

**RC2**: 'Comment on sp-2024-24', Anonymous Referee #2, 24 Oct 2024 reply

General comments:

This manuscript provides a brief overview of recent developments in numerical models for sea-ice. It covers fundamental physical processes, modelling approaches, and operational systems with data assimilation techniques, and provides a summary of modern numerical models and operational system for sea ice. The manuscript also draws attention to the challenges and concerns in developing sea ice modeling, numerical solvers, and machine learning applications. The manuscript is concise and informative, thus suitable for publication.

Specific comments:

Line 60: "Smaller ice floes offer more reflecting edges and are more efficient at attenuating waves." This statement is incomprehensive as it only mentions reflection or scattering, but not dissipation through multiple energy non-conservative processes in the ice effects on waves, which is also mentioned in Squire (2020).

*Thanks for the insightful remark, dissipation has been added.*

Technical corrections:

Regarding Chapter 2's organization: The "Overview of processes in sea ice" currently includes sections that aren't strictly processes (numerical models and data assimilation). I would

recommend that the author consider restructuring the technical sections into a new chapter regarding the current modeling approaches.

*Correct, all these subsections have been turned into sections.*

In Table 1, it mentions "*Output interpolated to 9 km" in the caption, but the corresponding entry isn't shown in the table. Moreover, the links for GIOPS and RIOPS are not precise, and Met Office FOAM may also need a relevant link.

*Thanks, the missing asterisk has been added to the ECMWF line in the table.*

*The links for GIOPS and RIOPS were provided by ECCC.*

*The FOAM high-resolution forecast system is distributed to selected users and has been removed from the table..*